# A pilot feasibility study of human-centered design for cirrhosis care: Development and pilot testing of SMARTLiver prototype, a FHIR-based clinical decision support system for hepatology

Keerthika Sunchu[1], Archita P. Desai [ID][2], Raj Vuppalanchi[2], Saptarshi Purkayastha [ID][1]*

**1** Department of Biomedical Engineering and Informatics, Indiana University Indianapolis, Indianapolis, Indiana, United States of America, **2** Division of Gastroenterology and Hepatology, Indiana University School of Medicine, Indianapolis, Indiana, United States of America

* saptpurk@iu.edu

## Abstract

Management of cirrhosis suffers from poor guideline adherence due to fragmented electronic health record (EHR) systems that scatter critical patient data across multiple modules, creating cognitive burden for clinicians and impeding evidence-based care delivery. We developed SMARTLiver, a Substitutable Medical Applications and Reusable Technologies on Fast Healthcare Interoperability Resources (SMART-on-FHIR) clinical decision support application employing human-centered design principles to consolidate patient data, incorporate evidence-based guidelines, and enhance cirrhosis care workflows. Following literature reviews of cirrhosis management guidelines and clinical workflow analysis within our health system, we created a FHIR-based application integrating automated task management, prognostic scoring, patient-reported outcomes, and real-time clinical decision support features. Usability evaluation with five clinical staff members using Think-Aloud protocols and the validated Health-ITUES survey revealed high satisfaction scores for Clinical Utility (4.4-4.6/5.0) and User Interface design (4.2/5.0), with moderate scores for workflow integration (4.0/5.0) and decision support (3.8-4.0/5.0). Qualitative feedback aligned with quantitative results, identifying enhancement opportunities in customization controls and notification management. The SMARTLiver prototype demonstrated technical feasibility in aggregating fragmented clinical data into a unified interface, automating evidence-based task generation, and maintaining interoperability across healthcare systems. This pilot study provides initial evidence for the potential of SMART-on-FHIR technology to address EHR fragmentation in cirrhosis care, though clinical effectiveness remains to be demonstrated.

which permits unrestricted use, distribution, and reproduction in any medium, provided the original author and source are credited.

**Data availability statement:** The datasets supporting the conclusions of this article have limited availability due to privacy and confidentiality considerations. Survey responses and think-aloud protocol transcripts underlying the usability evaluation findings cannot be shared due to the small number of participants (n=5), which creates risks of participant identification even with de-identification efforts. The institutional review board (IRB) protocol governing this study restricts sharing of individual-level qualitative data to protect participant confidentiality. Additionally, clinical data generated during application testing cannot be shared due to patient privacy restrictions under the Health Insurance Portability and Accountability Act (HIPAA). Researchers with specific questions about the study methodology or aggregate findings should contact the Indiana University IRB and Human Research Protection Program (HRPP) at irb@iu.edu.

**Funding:** The author(s) received no specific funding for this work.

**Competing interests:** The authors have declared that no competing interests exist. KS, APD, RV, and SP report no financial relationships with any organizations that might have an interest in the submitted work; no other relationships or activities that could appear to have influenced the submitted work.

## Author summary

We developed a computer application to help physicians and other healthcare providers better care for patients with severe liver disease, i.e., cirrhosis, a condition that affects millions worldwide and requires complex, coordinated treatment. Currently, healthcare providers (HCP) must click through many different screens in computer systems to gather all the information they need about a patient, which is time-consuming and can lead to important details being overlooked. Our team created an app that brings together all relevant patient information - laboratory results, medications, symptom reports, and treatment recommendations - into a single, easy-to-use interface. We worked closely with HCPs and nurses to understand their daily challenges and designed the app to fit naturally into their existing workflows. When we tested the app with five healthcare providers, they found it significantly improved their ability to access patient information and follow evidence-based treatment guidelines. The app automatically generates reminders for important medical tasks and helps HCPs track patient progress over time. The current work demonstrates how thoughtfully designed technology can address challenging problems in healthcare delivery. By reducing the burden of navigating complex computer systems, we hope to help HCPs spend more time focusing on patient care and ensure that patients with liver disease receive the comprehensive, guideline-directed treatment they need.

## Introduction

Fragmented clinical data represents the most significant barrier to effective chronic liver disease management, forcing clinicians to navigate multiple disconnected electronic health record (EHR) modules to gather essential patient information [1,2]. This fragmentation challenge is particularly acute in the management of complex chronic diseases like cirrhosis, a leading cause of morbidity, mortality and healthcare-related costs [3–5]. Due to the fragementation, providers must synthesize data from laboratory trends, imaging results, medication histories, and procedure notes scattered across 6–8 different screens, a cognitive burden that directly impedes timely, evidence-based decision-making. Despite the availability of evidence-based clinical practice guidelines from professional societies such as the American Association for the Study of Liver Diseases (AASLD), adherence to these recommendations remains disappointingly low in clinical practice due to cognitive burden, low patient engagement and complex data [6,7].

The EHR data fragmentation forces clinicians to navigate what researchers term "display fragmentation," where critical clinical information is distributed across disparate locations in the medical record, requiring excessive navigation and mental context-switching [8]. For complex conditions like cirrhosis, where providers must synthesize information from laboratory trends, imaging results, medication histories, and procedure notes across multiple organ systems, this fragmentation creates a significant cognitive burden that impedes timely, evidence-based decision-making [9].

To address these fragmentation challenges in cirrhosis management, we leveraged the SMART on FHIR framework (Substitutable Medical Applications and Reusable Technologies on Fast Healthcare Interoperability Resources), which enables standardized integration across different EHR systems [10]. This framework has demonstrated success in recent implementations, including integration of Patient-Reported Outcomes across multiple EHR platforms and emergency department clinical decision support [11].

## The promise and challenges of clinical decision support

Clinical Decision Support Systems (CDSS) represent a promising solution to bridge the gap between evidence and practice. The global CDSS market, valued at $5.3 billion in 2023, is projected to reach $14.9 billion by 2033, driven by increasing demand for quality healthcare and the growing complexity of clinical decisions [12]. These systems aim to provide healthcare professionals with patient-specific assessments and recommendations at the point of care, potentially reducing the cognitive burden of complex decision-making.

However, CDSS implementation in healthcare has yielded mixed results. While systematic reviews demonstrate that both commercially and locally developed CDSS can improve healthcare process measures across diverse settings, evidence for meaningful improvements in clinical outcomes, economic benefits, and provider efficiency remains sparse [13]. A key factor determining CDSS success appears to be provider involvement in the development process, suggesting that user-centered design approaches are critical for adoption and effectiveness [14].

In liver disease management specifically, CDSS have been underutilized despite the clear need. A recent human-centered design study for cirrhosis management found that clinicians identified "aggregation of relevant clinical data into one interface" as the most important feature for improving care delivery [9]. This finding underscores the fundamental challenge of EHR fragmentation in complex disease management.

## The SMART on FHIR revolution

A technological solution to the interoperability crisis has emerged through the SMART on FHIR framework (Substitutable Medical Applications and Reusable Technologies on Fast Healthcare Interoperability Resources). Developed to enable "write once, run anywhere" healthcare applications, SMART on FHIR provides a standardized platform that allows third-party applications to securely integrate with any compliant EHR system [15,16]. By 2024, the framework has become a regulatory requirement for ONC-certified health IT systems [17], mandating universal APIs for health information access [18,19].

Recent implementations demonstrate the framework's potential. A 2023 study successfully integrated Patient-Reported Outcomes Measurement Information System (PROMIS) data across 18 ambulatory care sites using three different EHR platforms, proving the feasibility of scalable, secure, and resource-efficient SMART on FHIR applications [20]. Similarly, emergency department clinical decision support tools for pneumonia management in a randomized trial have shown promise using SMART on FHIR architecture [21].

Notably, our development and testing occurred entirely within a Cerner-based EHR environment at Indiana University Health, marking a significant departure from the predominant focus on Epic implementations in SMART-on-FHIR and human-centered design literature. This Cerner-centric approach addresses an important gap, as most published FHIR applications have been developed and validated exclusively on Epic platforms, potentially limiting their generalizability to the substantial proportion of healthcare systems using alternative EHR vendors [22].

## Research aims and innovation

Our methodology employed an iterative human-centered design process, including clinical workflow observation, stakeholder interviews, prototype development, and usability testing with healthcare providers (detailed methods are presented at the end of this manuscript). Fig 9 illustrates our human-centered design framework, showing the progression from

initial clinical observation through iterative prototyping to final usability evaluation. To address the challenges of cirrhosis management through the iterative, human-centered design of SMARTLiver, a SMART on FHIR clinical decision support application specifically for cirrhosis management. Building on recent advances in interoperable health technology and human-centered design principles, we sought to:

1. **Design and develop** a SMART on FHIR application that consolidates fragmented clinical data for patients with cirrhosis and end-stage liver disease.

2. **Incorporate evidence-based guidelines** into actionable clinical tasks within an interoperable framework.

3. **Create intuitive data visualization features** including prognostic scoring and patient-reported outcomes integration.

4. **Evaluate usability and workflow impact** through mixed-methods assessment with healthcare providers.

This work represents one of the first applications of SMART on FHIR technology specifically for liver disease management, addressing the documented need for data aggregation and clinical decision support in hepatology care.

## Results

We conducted a comprehensive human-centered design process to understand the real-world challenges facing clinicians in cirrhosis management. Through full-day observation of the outpatient practice of hepatology providers at Indiana University Health (IUH), we systematically documented their workflow for cirrhosis care, paying particular attention to how they navigated the Cerner EHR system (Oracle Health, formerly Cerner). This observational study revealed several critical pain points that directly informed our application design.

### Human-centered design process and clinical workflow analysis

We conducted full-day observations of hepatology providers at Indiana University Health, systematically documenting their Cerner EHR navigation patterns and workflow challenges for cirrhosis management. The workflow analysis identified that providers routinely struggled with fragmented data access across multiple EHR modules. Essential clinical information for cirrhosis management—including laboratory trends, medication histories, procedure notes, and imaging results—was distributed across separate screens and interfaces, requiring excessive navigation and mental context-switching. We documented that clinicians needed to access an average of 6–8 different EHR screens to gather comprehensive information for a single cirrhosis patient encounter.

A significant finding was that several important clinical scoring systems were either not easily accessible or inadequately visualized within the existing EHR interface. Critical clinical scores for cirrhosis management, such as MELD (Model for End-Stage Liver Disease), Child-Pugh scores, Fibrosis-4 (FIB-4) index, APRI (AST to Platelet Ratio Index), and liver stiffness measurements (LSM), required manual calculation or were buried within clinical notes. Additionally, patient-reported outcome measures such as standardized symptoms or quality of life reports using measurement tools such as Patient-Reported Outcomes Measurement Information System (PROMIS-29 Profile) scores were not integrated into the clinical decision-making workflow, despite their importance for comprehensive patient assessment and providing patient-centered care. Fig 1 shows the clinical workflow that's enabled through the SMARTLiver application. Once this was completed, we moved on to iterative development cycles, which finally resulted in a usability study of SMARTLiver.

### Quantitative usability results

We administered the validated Health-ITUES survey to five clinical team members (2 RNs, 1 MA, 1 PA, 1 physician) following structured demonstrations of SMARTLiver's features. While SMARTLiver has not yet been integrated into routine hepatology practice at IUH, we demonstrated the SMARTLiver application and then measured its usability by

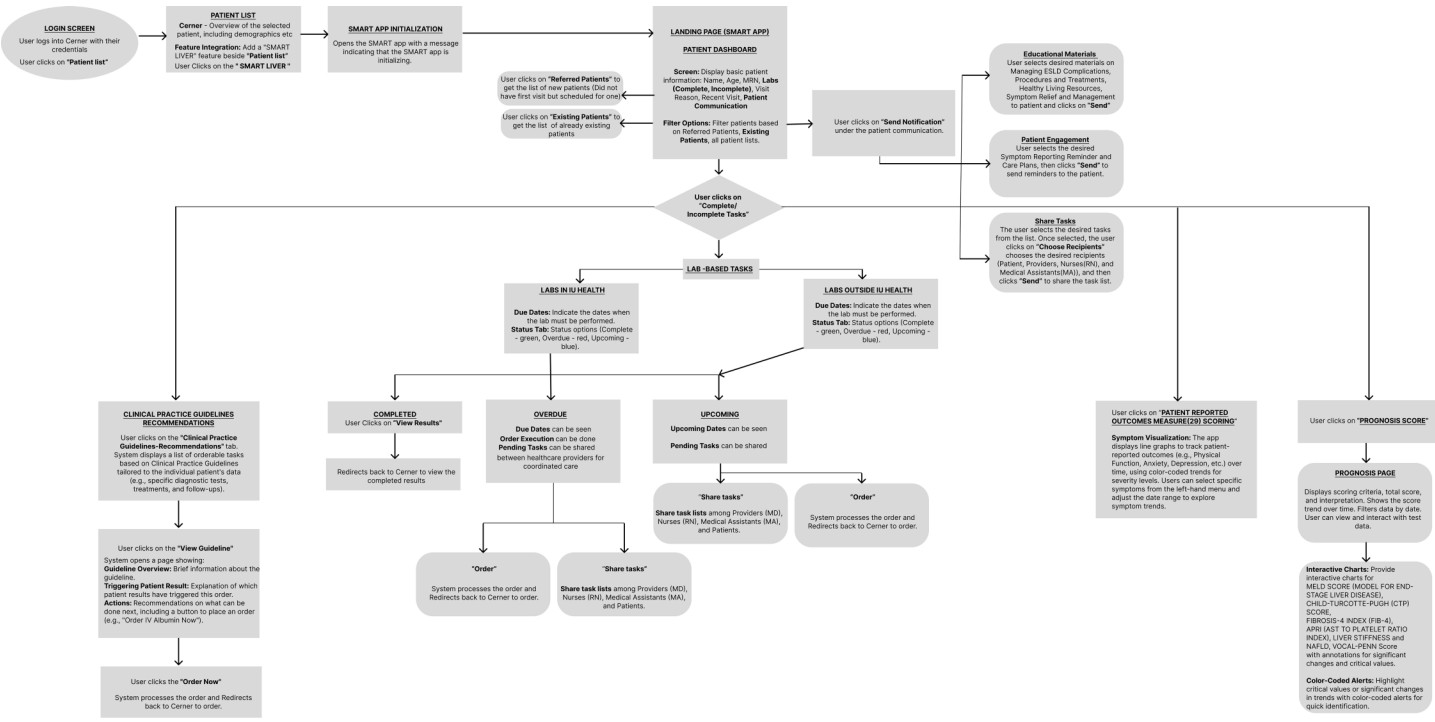

**Fig 1. Representation of the SMARTLiver clinical workflow.**

administering the Health-ITUES survey [23] to the clinical team members, who provided structured quantitative assessment of SMARTLiver's plausible usability across five critical domains for clinical decision support systems. We mapped survey responses to five constructs (Fig 2): Integration & Workflow, Clinical Utility, User Interface & Ease of Use, Decision Support & Patient Management, and Customization & Control.

**Clinical Utility** received the highest satisfaction scores, with mean ratings of 4.4-4.6 out of 5.0, indicating strong user perception of the application's clinical value. All participants agreed that the application provided valuable clinical functionality.

**User Interface & Ease of Use** also scored highly, with average ratings of 4.2 across relevant survey items. Eighty percent of participants agreed or strongly agreed that the interface was intuitive and easy to navigate. No participants expressed disagreement with interface usability statements.

**Integration & Workflow** received mixed responses: on the primary integration item, 2 participants (40%) strongly agreed, 2 participants (40%) agreed, and 1 participant (20%) remained neutral that the application integrated well into clinical workflows. When averaging across all Integration & Workflow survey items, the positive response rate was 80% (combining 'agree' and 'strongly agree' responses).

**Decision Support & Patient Management** showed positive ratings as well, with average scores of 3.9. Sixty percent of participants agreed or strongly agreed that the decision support features were effective, while 40% remained neutral and none disagreed. This pattern suggested that while decision support features were viewed positively, SMARTLiver had not yet achieved optimal integration into clinical practice.

**Customization & Control** received the lowest scores, ranging from 3.2-4.0, with only 60% of participants agreeing or strongly agreeing that they had adequate control over application settings and display options (10% strongly agreeing, 50% agreeing). Notably, 30% remained neutral and 10% disagreed, indicating this as the primary area requiring enhancement.

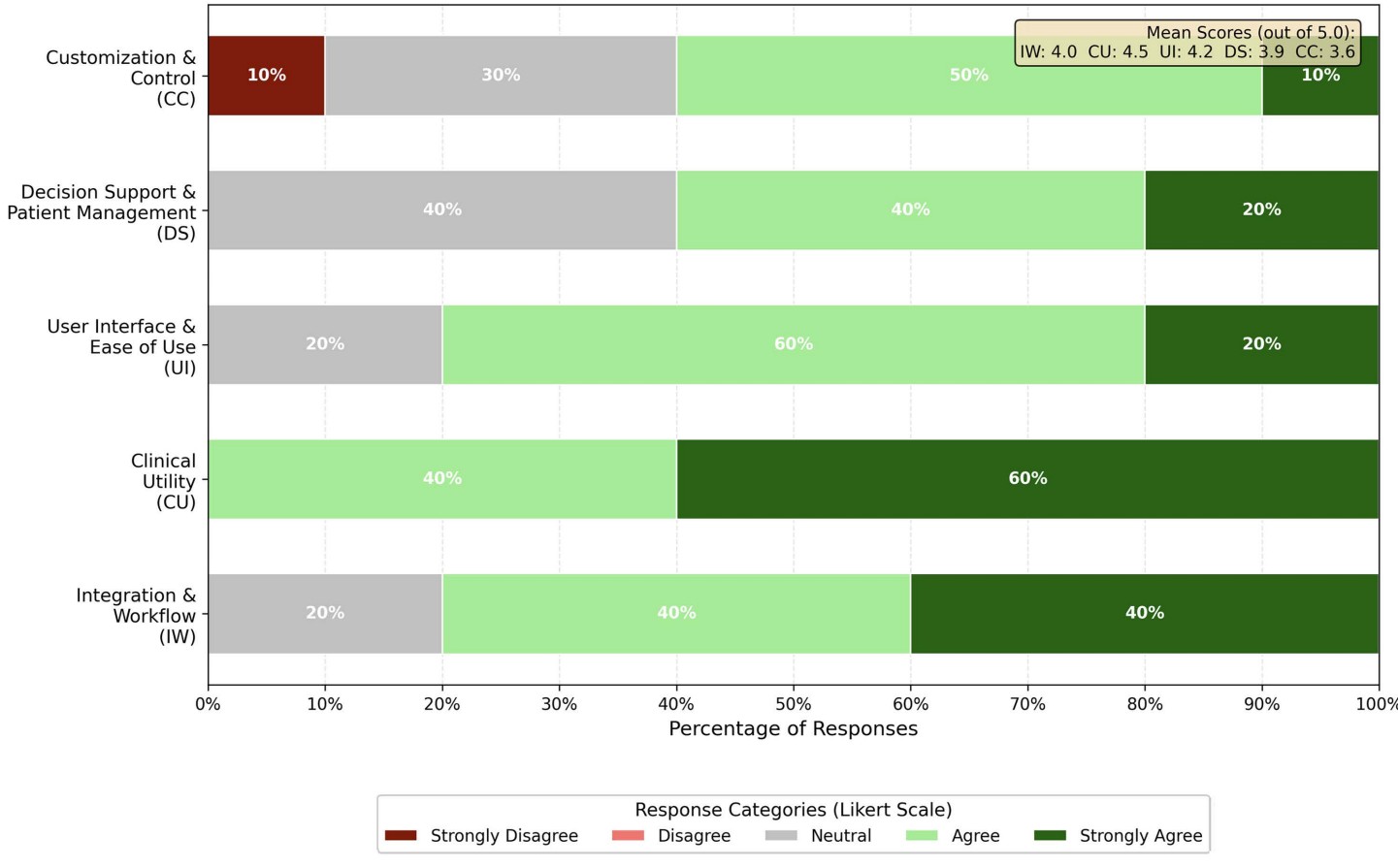

**Fig 2. Health-ITUES Usability Survey Results by Domain.**

## Qualitative feedback analysis

Using Think-Aloud protocols during task completion and affinity mapping of recorded feedback, we identified recurring usability themes across participant roles.

**Workflow Integration Insights**: Participants appreciated the patient filtering functionality, with multiple users noting that the distinction between "Existing" and "Referred" patients would "organize workflows and save time." However, they identified specific friction points, including inconsistent date formats across different data displays and intrusive automatic pop-up notifications that disrupted clinical tasks. One nurse specifically commented that standardized date formats would "reduce cognitive friction" when reviewing laboratory results.

**Clinical Utility Strengths**: The most enthusiastic feedback centered on clinical decision-making enhancements. Participants particularly valued the PROMIS-29 drill-down functionality, which enabled detailed viewing of patient-reported outcome trends over time and allowed comparison to age-adjusted national norms. Clinicians noted that having integrated risk scores "broadens the clinical picture," and highlighted that symptom tracking in subcategories would improve care planning precision.

**Decision Support Gaps**: While participants acknowledged the value of embedded clinical decision support content, they identified important limitations. A common theme was the need for enhanced task sharing and coordination mechanisms. One nurse noted that the ability to assign tasks to other team members would "enhance team-based care."

Participants also requested clearer notification confirmations to ensure that when clinical alerts were addressed, the entire care team would be informed to prevent task duplication.

**Customization Requests**: The strongest feedback theme involved requests for greater user control over interface elements and workflows. Participants consistently requested patient age display prominently in patient lists, ability to configure dropdown menus based on individual preferences, and a notification control panel for managing alerts. One participant summarized the sentiment as wanting "more control over what the app shows and when."

**Interface Refinements**: While overall interface feedback was positive, participants suggested specific improvements including tooltips for medical abbreviations to reduce confusion for new users, and provision of normal laboratory value ranges in dropdown menus for quick reference. A nurse mentioned that having reference ranges visible would "add clarity when reviewing labs."

## Feature prioritization and implementation

Based on systematic analysis of both quantitative survey results and qualitative feedback themes, we prioritized a comprehensive set of application improvements designed to address the most significant usability gaps while reinforcing existing strengths.

**Patient List Enhancements**: We refined the patient filtering logic to provide clearer categorization, defining "Existing" patients as those with at least one prior visit and "Referred" patients as those with none. Patient age was added to the dashboard display to address the most frequent customization request. These changes directly responded to workflow integration feedback and provided the demographic context that clinicians indicated was essential for rapid patient assessment, as can be seen in Fig 3.

**Laboratory Results Organization**: To address confusion about duplicate testing and improve clinical efficiency, we segregated laboratory results by source, clearly separating IU Health internal results from external laboratory data (see Fig 4). This change eliminated the need for providers to verify whether tests had already been completed within their system. Additionally, we standardized all date formats in the application to the "Month Day, Year" format, addressing the cognitive friction participants reported when reviewing clinical data.

**Enhanced PROMIS-29 Integration**: Building on the positive feedback about patient-reported outcomes, we implemented an interactive drill-down system for PROMIS-29 Profile domain and summary scores (see Fig 5) and provided visual guidance on interpreting the scores (normal, mild, moderate, severe). Clinicians could now click on summary scores to access detailed trajectory analysis (see Fig 6) and domain scores over multiple visits while easily interpreting the scores compared to national normative data. This longitudinal visualization capability enabled monitoring of symptom trends and quality-of-life changes, directly supporting the personalized care planning that participants indicated was crucial for cirrhosis management.

**Team-Based Care Workflow**: We developed a comprehensive task assignment system to facilitate care coordination, the most frequently requested enhancement. Providers could now assign follow-up tasks to shared nurse or medical assistant pools, with clear provenance tracking showing who requested each task (see Fig 7). The system also implemented confirmation messaging to reassure users that clinical actions had been recorded and communicated to the appropriate team members, addressing concerns about task completion transparency.

**Risk Assessment Expansion**: We incorporated the VOCAL-Penn risk score into the prognostic assessment panel, responding to specific clinician requests for additional evidence-based prognostic tools (see Fig 8). Within the scoring interface, we added toggle controls for normal reference ranges, allowing users to show or hide these values based on their current needs and experience level.

**Patient Education Customization**: The patient education system was enhanced to support targeted content delivery based on clinical relevance. Education materials were organized into selectable categories (cirrhosis complications, procedures, healthy living, symptom relief) with specific subtopics, enabling providers to send personalized educational content

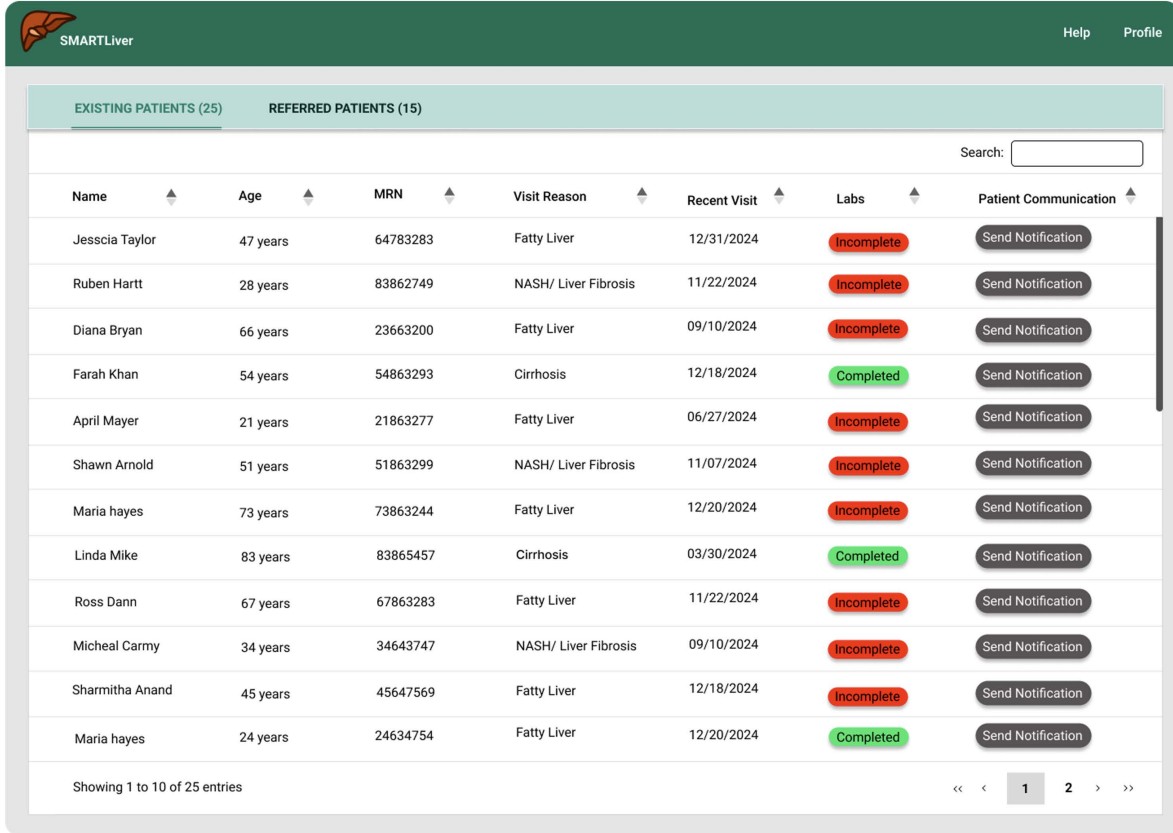

**Fig 3. Patient list summarizing updates from labs.**

rather than generic materials. The patient engagement section was expanded to include structured symptom reporting categories (general, gastrointestinal, quality of life) along with free-text options for comprehensive symptom tracking.

These prioritized improvements directly addressed the lower-scoring aspects of usability assessment (workflow integration, decision support effectiveness, and customization control) while preserving and enhancing the application's clinical utility and interface strengths that received high user satisfaction scores.

## Discussion

This pilot study provides initial evidence that SMART on FHIR technology may help address EHR fragmentation in specialty care settings. Our preliminary usability testing with five participants offers early insights that raise questions about existing assumptions in healthcare IT implementation, though larger studies are needed to validate these observations.

### Major conclusions and clinical significance

**First, our preliminary findings suggest that data aggregation may be more valued by clinicians than algorithmic sophistication in this context**. Contrary to prevailing assumptions that advanced clinical decision support requires complex automated inference and machine learning capabilities, our evaluation revealed that clinicians most valued the simple aggregation of relevant clinical data into a unified interface. This finding fundamentally challenges the field's focus on developing increasingly sophisticated automated reasoning systems. The 100% positive response rate for clinical utility

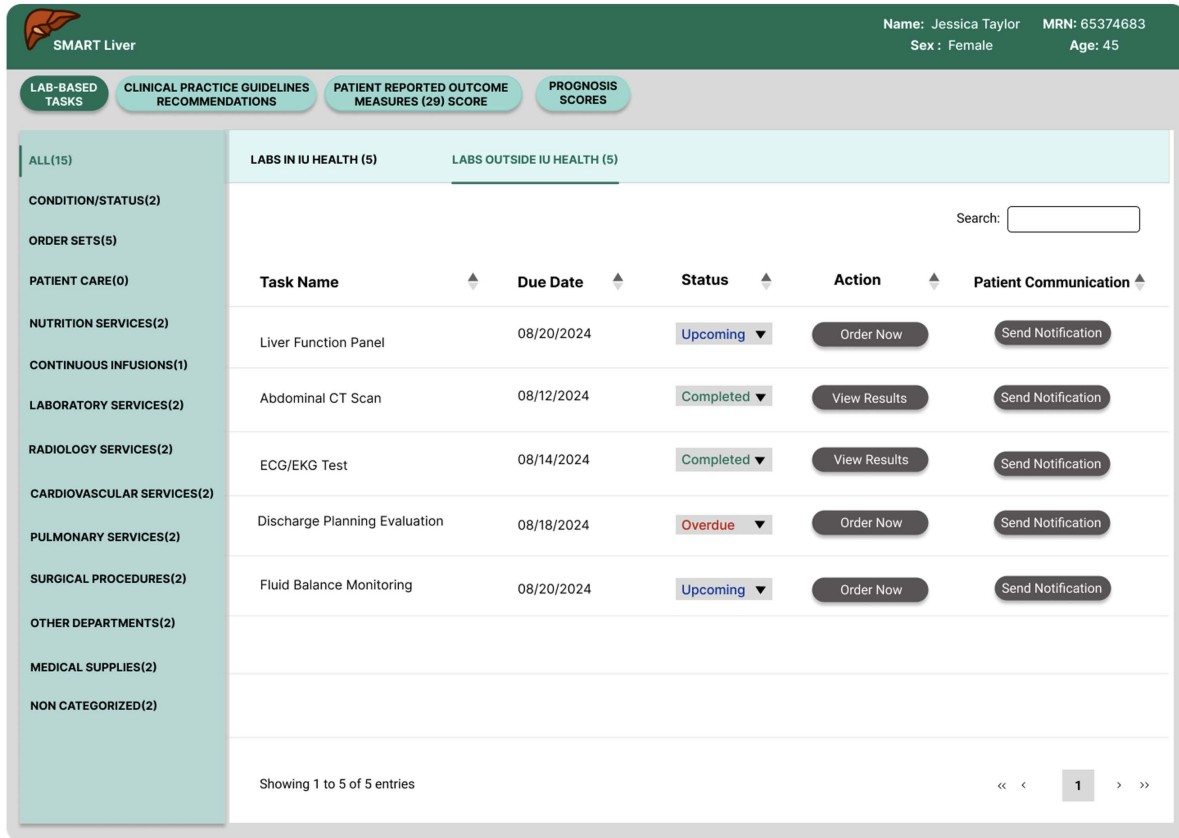

**Fig 4. Organizing lab results, showing internal or external labs.**

(mean scores 4.4-4.6/5.0) occurred primarily because SMARTLiver eliminated the cognitive burden of navigating 6–8 separate EHR screens, not because of advanced decision support algorithms. This suggests that the healthcare IT field has been solving the wrong problem—investing in algorithmic complexity while the real barrier to evidence-based care remains information fragmentation.

**Second, our limited usability testing indicates that workflow integration trumps technical interoperability for clinical adoption**. While we successfully achieved technical interoperability across multiple FHIR implementations (Cerner R4, Xealth R4), clinical adoption barriers centered on workflow disruption rather than technical functionality. The moderate scores for workflow integration (4.0/5.0) and customization control (3.2-4.0/5.0) indicate that technical standards compliance is necessary but insufficient for clinical success. This finding challenges the field's assumption that establishing technical interoperability standards will automatically improve clinical workflows. Instead, our results suggest that significant additional investment in user experience design and workflow customization capabilities is required to realize the benefits of interoperable systems.

**Third, our prototype demonstrates technical feasibility for specialty-specific clinical decision support across EHR vendors, though scalability remains to be validated**. Our successful implementation across multiple FHIR implementations with vendor-specific customizations demonstrates that the "write once, run anywhere" vision of SMART on FHIR is technically achievable for complex medical domains. However, this requires substantial mapping and abstraction work to handle vendor variations—a finding that has significant implications for the economic viability and scalability of interoperable clinical applications.

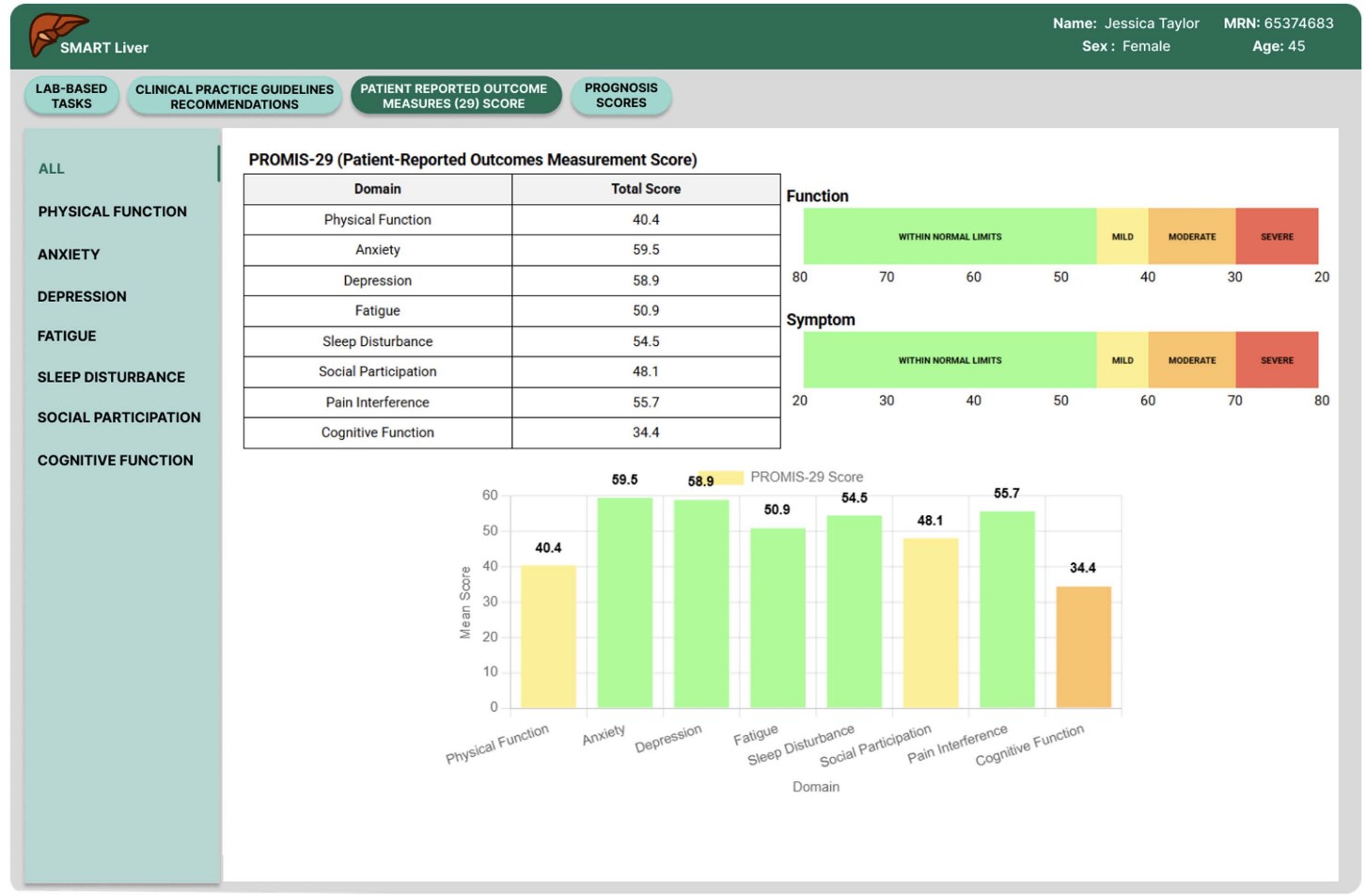

**Fig 5. Summary of PROMIS-29 scores per domain.**

## Impact on health informatics field assumptions

These preliminary observations from our pilot study raise questions about several prevailing assumptions and show that addressing basic information access problems may yield greater clinical value than developing advanced automated reasoning capabilities. Similarly, the field has operated under the assumption that technical interoperability standards like FHIR would naturally translate into clinical workflow improvements. Our mixed results for workflow integration demonstrate that this assumption oversimplifies the complex sociotechnical challenges of healthcare IT implementation. Technical standards enable interoperability but do not guarantee usable, adoptable solutions. Here, human-centered design approach with iterative user-engagement can lead to higher fidelity in user adoption and workflow integration.

Our successful implementation in a Cerner environment is particularly noteworthy given the health informatics field's heavy emphasis on Epic-based SMART-on-FHIR development. This demonstrates the framework's vendor-agnostic potential while highlighting the additional complexity required for non-Epic implementations.

The healthcare IT industry has also assumed that generic, one-size-fits-all clinical decision support systems would be more economically viable than specialty-specific solutions. Our successful development of a cirrhosis-specific application

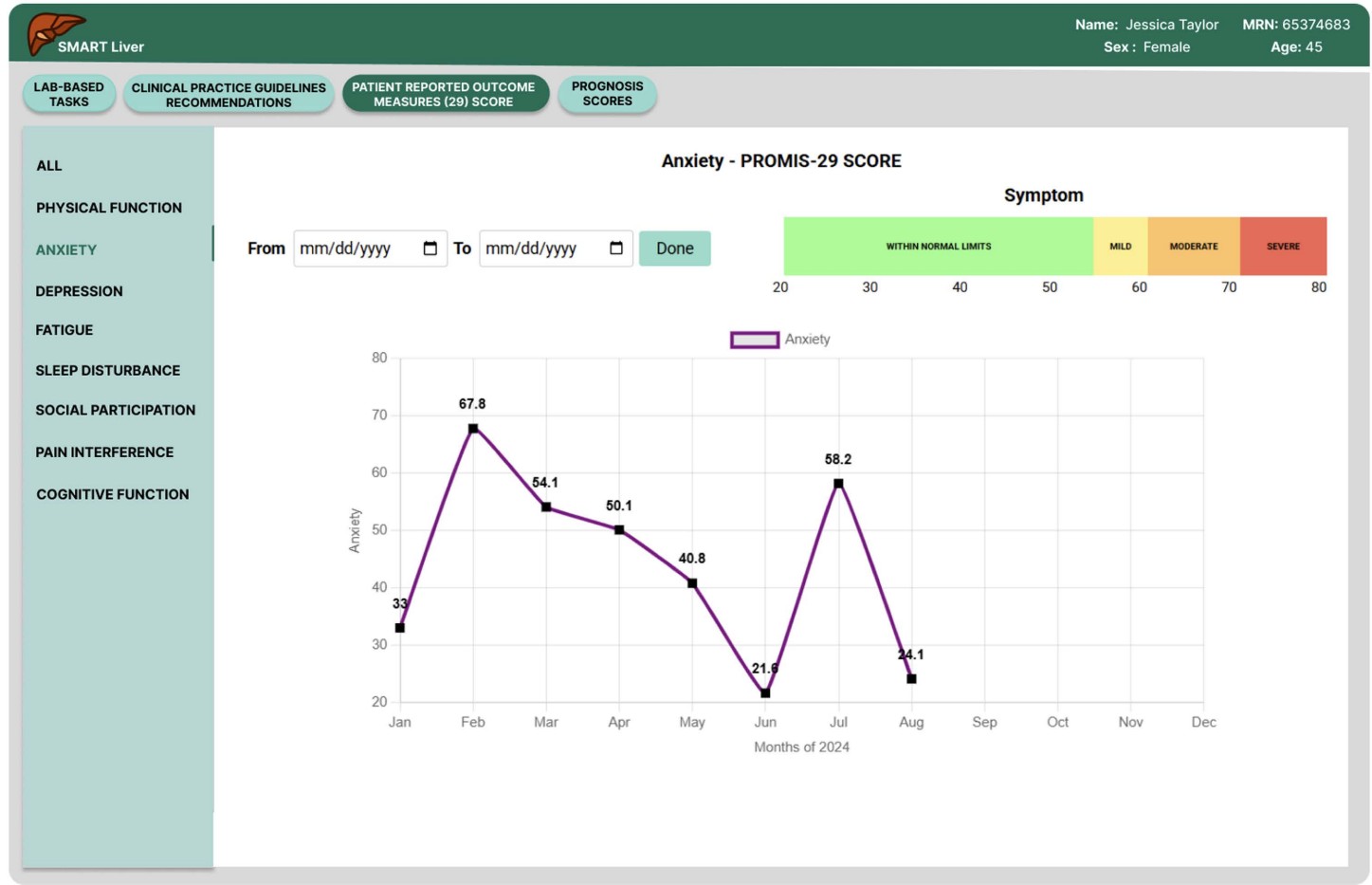

**Fig 6. PROMIS-29 patient trajectory over time.**

challenges this assumption, suggesting that targeted clinical applications may achieve higher clinical utility and adoption rates than generic systems, potentially justifying the additional development investment.

## Future research directions

Implementation Continuity Considerations: Indiana University Health is scheduled to transition from Cerner to Epic in approximately two years (2027), which presents both challenges and opportunities for SMARTLiver's evolution. While this transition will require remapping our FHIR resource implementations to Epic's specifications, it also offers a unique opportunity to validate SMARTLiver's cross-vendor portability - a critical test of SMART-on-FHIR's 'write once, run anywhere' promise. We plan to leverage this transition as a natural experiment, documenting the effort required for cross-platform migration and identifying which components remain stable versus those requiring vendor-specific adaptation. For our planned randomized controlled trial, we will either: (1) accelerate trial completion within the Cerner environment before the transition, or (2) design the trial with a planned interim analysis coinciding with the EHR migration, allowing us to assess the application's performance across both platforms. This real-world vendor transition will provide valuable insights for the broader health informatics community regarding the true portability of SMART-on-FHIR applications.

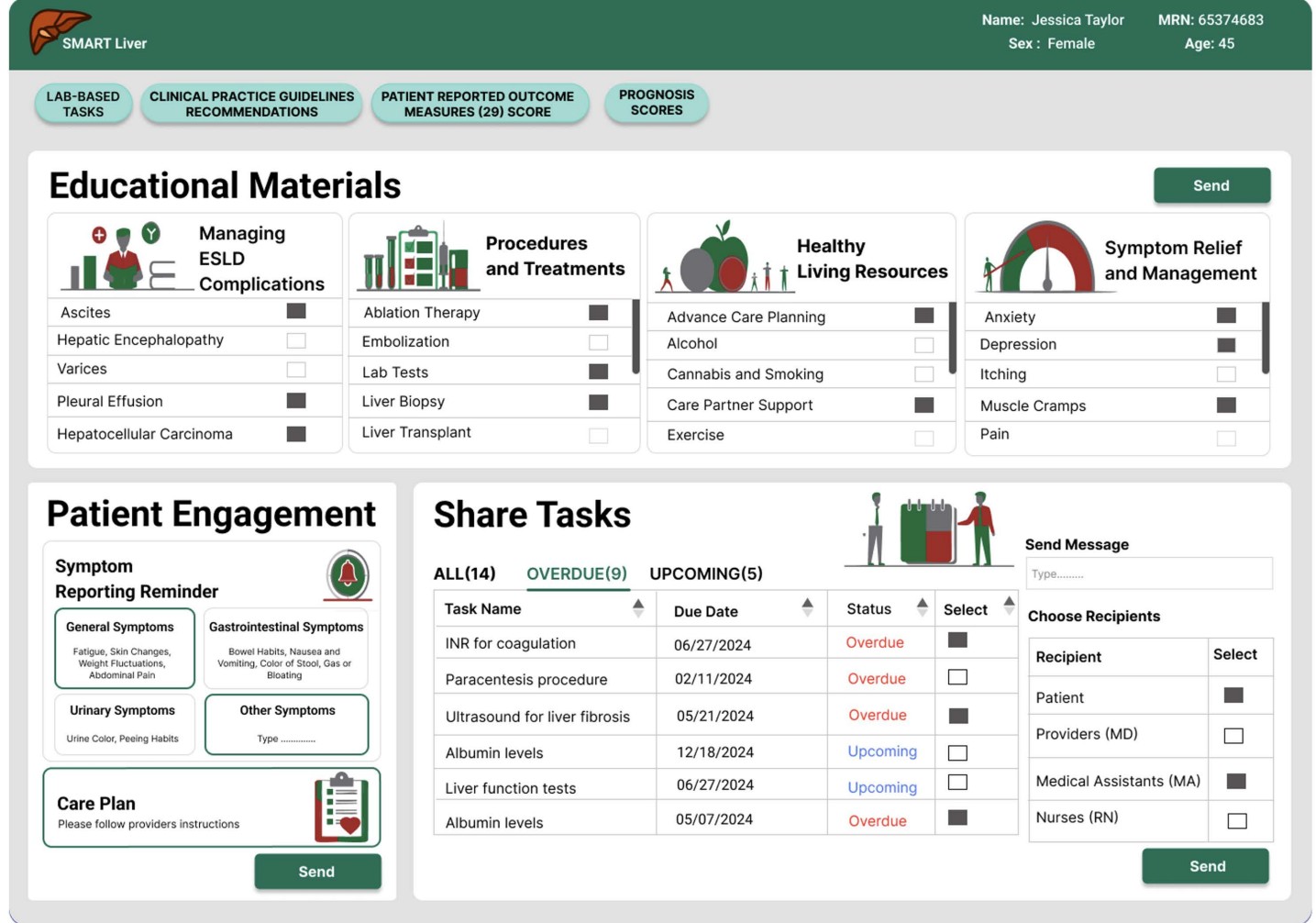

**Fig 7. Team-based workflow for patient education and shared tasks.**

Our study establishes the foundation for several critical research directions that could transform clinical decision-support implementation and evaluation. **The most immediate priority is a randomized controlled trial comparing SMARTLiver-assisted care to usual care for cirrhosis management**. While our usability evaluation demonstrates high clinician satisfaction and technical feasibility, clinical effectiveness remains unproven. Such a trial should measure quality outcomes including guideline adherence rates, time to critical interventions, patient-centered outcomes such as change in health-related quality of life, quality of life, symptom control, and satisfaction with care as well as long-term clinical outcomes such as hospitalization rates and disease progression.

**Longitudinal studies of provider workflow impact and burnout prevention** could address one of healthcare's most pressing challenges. If SMART on FHIR applications can demonstrably reduce cognitive burden and improve clinical efficiency, this could provide a compelling business case for widespread adoption. Such studies should measure objective workflow metrics (time per patient encounter, documentation burden, task completion rates) alongside subjective measures of provider satisfaction and burnout.

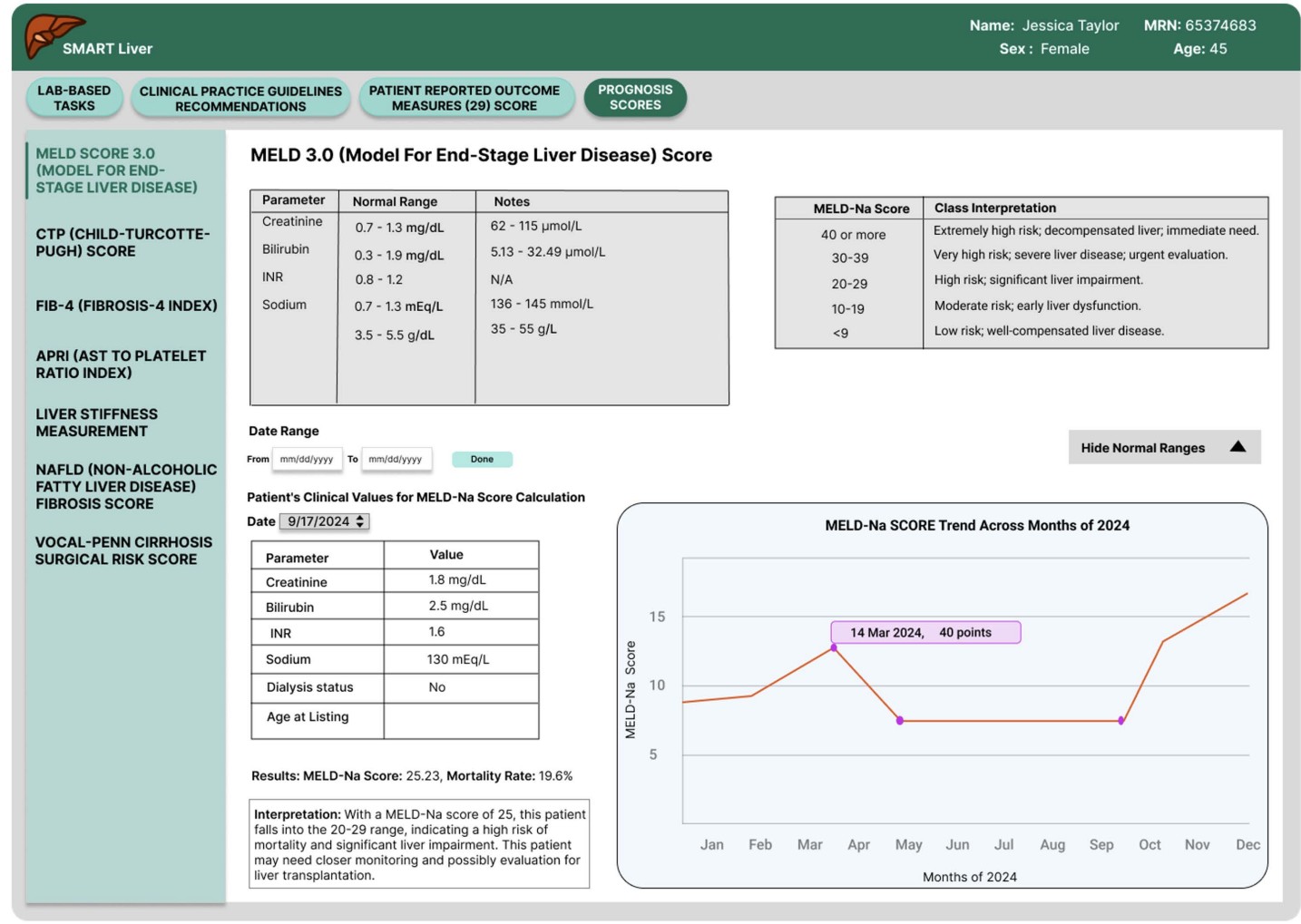

**Fig 8. A risk assessment dashboard with different liver-disease scoring.**

**Multi-site deployment studies across different EHR vendors and healthcare systems** represent another critical research direction. Our single-site evaluation cannot determine whether the technical and workflow solutions we developed will generalize to other environments. Understanding the resources required for deployment across different technical environments would inform economic viability assessments and vendor ecosystem development strategies. Integration within platforms such as Xealth® which cross multiple EHR environments may address these perceived barriers.

**Comparative effectiveness research across different medical specialties** would determine whether our findings generalize beyond hepatology. Different specialties have varying degrees of guideline complexity, workflow standardization, and data fragmentation. Understanding which specialties would benefit most from SMART on FHIR clinical decision support could guide implementation priorities and resource allocation.

## Limitations and future development priorities

Our study has several limitations that constrain the generalizability of findings and highlight priorities for future development. The single-site evaluation with five participants, while sufficient for initial usability assessment, cannot predict

adoption patterns across diverse clinical environments. We also did not assess clinical effectiveness, which remains the ultimate test of clinical decision support value.

Our focus on cirrhosis management, while providing deep domain expertise, may not generalize to other specialties with different workflow patterns, guideline complexity, or data requirements. Additionally, our evaluation occurred in a controlled environment with motivated participants, which may not reflect real-world adoption challenges including competing priorities, workflow interruptions, and organizational resistance to change.

**Data Completeness Considerations**: An important limitation of the current SMARTLiver implementation is its reliance on structured FHIR data elements, which may not capture all clinically relevant information. Laboratory results, clinical observations, and other critical data often exist in free-text notes, scanned documents from external providers, or PDF attachments that fall outside the current application's extraction capabilities. For instance, referring providers' assessment letters, outside hospital discharge summaries, and historical paper-based records - all potentially containing vital cirrhosis management information - remain inaccessible through standard FHIR queries. While SMARTLiver represents a substantial advancement in consolidating multiple system data streams, future iterations should explore natural language processing and optical character recognition technologies to extract information from unstructured sources, thereby providing a more complete clinical picture for decision support.

**Technical limitations** include the substantial vendor-specific customization required despite FHIR standardization, which raises questions about the economic sustainability of multi-vendor application development. Performance characteristics under realistic clinical loads remain unknown, and security considerations for multi-application FHIR ecosystems require further investigation.

**Future development priorities** should address these limitations through expanded clinical evaluation, multi-specialty application development, and infrastructure optimization for scaled deployment. The field needs standardized evaluation frameworks for SMART on FHIR applications, economic models for sustainable application ecosystems [24,25], and technical architectures that support hundreds of concurrent clinical applications without compromising performance or security.

## Materials and methods

### Human-centered design approach

Our development process followed a human-centered design approach, outlined in Fig 9, focusing on understanding user needs, creating solutions that address these needs, and iteratively refining the design based on user feedback.

### Literature review on cirrhosis management

A targeted literature review was conducted to inform the design of a SMART on FHIR application for cirrhosis management. Key patient variables were drawn from recent PubMed-indexed reviews, highlighting common etiologies (alcohol-related liver disease, metabolic dysfunction-associated steatotic liver disease, hepatitis C) and frequent symptoms (muscle cramps, pruritus, poor sleep, sexual dysfunction) associated with cirrhosis. In parallel, AASLD (American Association for the Study of Liver Diseases) guidelines were reviewed to integrate evidence-based recommendations for cirrhosis care. AASLD quality measures further defined essential care processes and clinical metrics – covering the full spectrum of cirrhosis management from ascites and variceal bleeding to hepatocellular carcinoma (HCC) surveillance and timely transplant evaluation. Together, these sources ensured that the app's clinical content (e.g., recommended interventions like diagnostic paracentesis for ascites-related hospitalizations or early antibiotic prophylaxis for variceal hemorrhage) aligns with established standards.

Additionally, patient-focused resources from https://cirrhosiscare.ca were incorporated to emphasize education and symptom self-management. These materials provided accessible guidance on key complications of cirrhosis, daily

## Human-Centered Design Process for SMARTLiver Development

**Fig 9. The Human Centered Design process followed in SMARTLiver development.**

management tips, and warning signs for patients and caregivers. Finally, contemporary AASLD practice guidelines on managing cirrhosis complications (ascites, hepatic encephalopathy, portal hypertensive bleeding, etc.) were analyzed to ensure the app's decision support features reflect current best practices – for example, recognizing new-onset ascites as a trigger for transplant referral and palliative care involvement in advanced disease. In summary, this comprehensive review of literature and guidelines was undertaken to guide the app's design and ensure its clinical relevance, grounding its features in up-to-date, evidence-based care standards for cirrhosis.

### Evaluation methods

**Participants.** We recruited five clinical staff participants from IU Health to evaluate the SMARTLiver application. The cohort included two registered nurses (RNs), one medical assistant (MA), one physician assistant (PA), and one physician provider (hepatology specialist). This mix was intentionally selected to capture diverse clinical perspectives across the care team. Nurses offered insights on care coordination and routine nursing workflows, the medical assistant provided feedback on front-line data entry and clinic processes, the physician assistant represented mid-level provider use of the tool, and the physician brought the specialist's viewpoint on decision-making in cirrhosis care. By involving multiple roles, the evaluation ensured that the system's usability and usefulness were assessed from all relevant angles of clinical practice, increasing the generalizability of the feedback.

**Interview process.** Each participant underwent a 45-minute usability session comprised of four structured segments:

- Introduction (5 minutes): We introduced the study and the SMARTLiver app, obtained verbal consent, and briefed the participant on the session format. Participants were instructed in the Think-Aloud protocol that is to continuously verbalize their thoughts and reasoning while using the app. This introduction established rapport and clarified that the focus was on evaluating the software (not the user's performance).

- Think-Aloud Usability Tasks (15 minutes): Participants completed a series of 10 representative tasks in the SMARTLiver app while narrating their thought process. The Think-Aloud methodology was used to capture real-time feedback on usability: as users verbalized what they liked, disliked, or found confusing, we noted any usability issues or positive

reactions. This approach provided rich qualitative data and immediate insight into the users' mental models and work-flows. We refrained from assisting, intervening only with reminders to "keep thinking aloud" if the participant fell silent, per standard usability testing practice. All interactions and comments were recorded via notes (and audio when permitted) for subsequent analysis.

- Feature-Specific Q&A (10 minutes): After the tasks, we conducted a brief semi-structured interview to probe specific features and observations. Participants were asked targeted questions about particular app components. This dialogue allowed clarification of Think-Aloud observations and invited participants to suggest improvements or elaborate on any confusion. The combination of Think-Aloud and follow-up questions ensured both spontaneous reactions and reflective feedback were captured.

- Health-ITUES Survey (12 minutes): In the final segment, participants completed the Health Information Technology Usability Evaluation Scale (Health-ITUES) questionnaire. This is a validated 20-item Likert-scale survey instrument assessing user perceptions of system usability across 5 key domains. For our study, we mapped the questions to five constructs relevant to clinical decision support: Integration & Workflow, Clinical Utility, User Interface & Ease of Use, Decision Support & Patient Management, and Customization & Control. Participants rated their agreement with statements (1 = Strongly Disagree to 5 = Strongly Agree) about SMARTLiver's usability. This quantitative step provided a structured measure of usability to complement the qualitative insights.

The overall interview process was thus a mixed-methods evaluation, combining observational usability testing (Think-Aloud) with focused interviewing and a standardized survey. This structure ensured a comprehensive understanding of the system's performance: the Think-Aloud yielded immediate, nuanced feedback on user experience, while the post-task survey quantified usability in a way that could be compared across participants and against benchmarks. By the end of each session, detailed notes, participant quotations, and survey responses were collected for analysis.

**Qualitative analysis method - affinity mapping.** All qualitative feedback from the sessions was synthesized using an affinity mapping approach. We transcribed or summarized each significant feedback point onto a virtual "sticky note" on a Miro board (a collaborative online whiteboard). To maintain participant context, each participant was assigned a unique color for their notes. This color-coding preserved the origin of feedback while allowing us to visually mix and group similar comments across different users. The team (including the facilitators and two faculty mentors) independently reviewed the notes and then collaboratively clustered related feedback into broader categories. For example, comments about "too many pop-up alerts" and "needing confirmation for notifications" were grouped into a theme of Notification Management, while notes like "want patient age displayed" and "ability to customize dashboard" fell under Customization needs. We iteratively rearranged and merged clusters until consensus was reached on a set of recurring themes that captured the essence of usability issues and improvement suggestions. Each cluster was given a descriptive label (theme name), and we noted how many participants (and which roles) contributed to each theme. This process highlighted which usability concerns were common across multiple users versus those voiced by only one role. The affinity mapping allowed us to see at a glance, for instance, that workflow integration issues (like patient filtering and date formats) had notes from nearly all five participants, indicating a high-priority theme. In total, this exercise yielded a set of major themes corresponding to the areas covered by the app's features and the survey constructs. By organizing feedback in this manner, we were able to identify patterns and prioritize issues in a systematic way. This ensured that our subsequent decisions for improvements were grounded in user-driven evidence rather than anecdotal impressions. Notably, the themes that emerged from affinity mapping aligned closely with the predefined survey domains (IW, CU, UI, DS, CC), which facilitated an integrated analysis of qualitative and quantitative data in the next step.

### Technical development

**FHIR resources and integration.** To ensure SMARTLiver could integrate smoothly with different systems, we compared FHIR R4 standards with both Cerner R4 and Xealth R4 implementations. Since IU Health uses Cerner,

understanding its FHIR structure was essential for real-time data access. At the same time, Xealth◇ ([https://www.xealth.com](https://www.xealth.com)), another platform partnered with IU Health - had its own version of FHIR. We used HL7 FHIR R4 as the reference standard and analyzed how Cerner and Xealth differed in terms of resource structure, required fields, and extensions.

This comparison helped us identify where vendor-specific customizations existed and what adjustments were needed to ensure compatibility. For example, Xealth◇ often used custom extensions and omitted certain fields, while Cerner followed a more restricted subset of FHIR resources (see S1 Data). By studying their documentation and example payloads, we mapped these differences and built handling logic to make the data interoperable. This process was necessary for SMARTLiver to reliably pull and interpret data across systems while staying aligned with FHIR R4 standards.

**Development environment and tools.** The SMARTLiver application was developed with Node.js as the runtime and Yarn for package management. Webpack was used to bundle and optimize modular JavaScript files corresponding to individual application features. The frontend utilized Bootstrap CSS for responsive design and Chart.js with the ChartDataLabels plugin for visualizing patient-reported outcomes and prognostic scores. SMART-on-FHIR integration was implemented using the fhirclient.js library, with OAuth2 and PKCE handling secure authentication against FHIR servers. Firebase Cloud Messaging and Firestore were used to send and store clinical notifications, with localStorage as a fallback in offline scenarios (see S2 Data). Patient context was maintained across modules using URLSearchParams, enabling dynamic rendering of personalized data across standalone HTML pages.

**UI design.** The user interface was prototyped in Figma (see S1 Fig), a cloud-based design tool that enables collaborative creation of interactive UI mockups. The design emphasizes intuitive navigation and consistent styling across all screens, aligning with established usability principles (e.g., maintaining internal and external consistency to minimize users' cognitive load. A standardized design system (colors, typography, and components) was applied to ensure a cohesive look and feel, so that users do not have to relearn interface conventions between screens. Layouts are responsive and optimized for various device sizes, following best practices in responsive design to ensure the interface adapts seamlessly from desktops to mobile devices. The interface content is organized around clinical workflow needs: the landing page provides filter tabs for Existing Patients (patients with prior visits in the system) and Referred Patients (new referrals not yet seen). Within each category, patient listings display key data fields relevant to care coordination – for example, age, medical record number (MRN), reason for visit, pending lab status, and referral source – allowing coordinators to quickly triage and navigate to the appropriate patient record. This Figma-designed layout with clear navigation and responsive elements was iteratively refined to maximize usability and consistency across the application.

## Conclusions

The SMARTLiver prototype demonstrates proof-of-concept for applying SMART on FHIR technology to cirrhosis management. Through human-centered design principles and iterative development, we have created an initial prototype that shows promise for addressing clinical data fragmentation while maintaining interoperability across different healthcare IT systems. Our pilot usability evaluation with five participants provides early feedback on potential workflow integration, though clinical effectiveness and broader adoption remain to be evaluated through larger studies.

## Supporting information

**S1 Fig. SMARTLiver Figma mockups.**
(PDF)

**S1 Data. FHIR R4 Resources from Cerner for SMARTLiver.**
(DOCX)

**S2 Data. Firebase and FCM implementation.**
(DOCX)

## Acknowledgments

We extend our sincere gratitude to the clinical staff participants from IU Health who generously contributed their time and expertise to evaluate the SMARTLiver application. We also acknowledge the research assistants who assisted with participant recruitment. Special recognition goes to Chandra Vikram for developing Carepal, the innovative patient-facing mobile application that connects seamlessly with SMARTLiver.

## Author contributions

**Conceptualization:** Keerthika Sunchu, Archita P. Desai, Saptarshi Purkayastha.

**Data curation:** Keerthika Sunchu.

**Formal analysis:** Keerthika Sunchu, Saptarshi Purkayastha.

**Funding acquisition:** Raj Vuppalanchi.

**Methodology:** Keerthika Sunchu, Saptarshi Purkayastha.

**Project administration:** Archita P. Desai, Saptarshi Purkayastha.

**Resources:** Archita P. Desai.

**Software:** Keerthika Sunchu, Raj Vuppalanchi.

**Supervision:** Archita P. Desai, Raj Vuppalanchi, Saptarshi Purkayastha.

**Validation:** Keerthika Sunchu, Archita P. Desai.

**Writing – original draft:** Keerthika Sunchu.

**Writing – review & editing:** Archita P. Desai, Raj Vuppalanchi, Saptarshi Purkayastha.

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
