## [Decision Letter · Decision Letter 0]

3 Sep 2025

Response to Reviewers
Revised Manuscript with Track Changes
Manuscript
**Journal Requirements:**
**Additional Editor Comments (if provided):**

Reviewer #2:

**Reviewers' Comments:**

**Comments to the Author**

1. Does this manuscript meet PLOS Digital Health’s publication criteria?

Reviewer #1: Yes

Reviewer #2: Partly

Reviewer #3: Yes

2. Has the statistical analysis been performed appropriately and rigorously?

Reviewer #1: N/A

Reviewer #2: N/A

Reviewer #3: Yes

3. Have the authors made all data underlying the findings in their manuscript fully available (please refer to the Data Availability Statement at the start of the manuscript PDF file)?

Reviewer #1: Yes

Reviewer #2: No

Reviewer #3: Yes

4. Is the manuscript presented in an intelligible fashion and written in standard English?

Reviewer #1: Yes

Reviewer #2: Yes

Reviewer #3: Yes

Reviewer #1: This manuscript by Sunchu et al describes the human centered design of a cirrhosis management CDS application. While the specific concept had been previously demonstrated in EPIC environments, this one is notable in that the background EHR is on the Cerner system. Specific comments below:

1. The introduction would benefit from it being more focused on the issue with chronic liver disease management. Fragmented data is the biggest challenge identified in prior studies and would highlight here as the major issue to be solved.

2. Historical details on the history and evolution of the EHR, while appreciated, are not relevant to the clinical problem or unmet need here. Recommend to get rid of the sections on fragmentation and information blocking.

3. The structured methods for the paper are presented at the end, please note this in the “Research Aims and Innovation” section as some of the results read like methods. A concept figure or diagram of the HCD exercises done would be helpful here.

4. Would revise the results section and sub-sections so that they contain a quick one sentence summary of what was actually done prior to diving into the results – it became very confusing to read since the methods are at the end of the paper.

5. Please reorder the color blocks in order of the Likert Scale, and legend for the stacked barchart figure. Please assign a numeric label for this figure with a label.

6. Under the integration and workflow survey results, how was a percentage of 73% participants obtained from a sample of n=5?

7. Was there any evaluation given to the completeness of the displayed information? For example with respect to laboratory data, data are often included in free text notes or scans from referring providers outside of the system. While the current application certainly represents a step forward in multiple system integration, there still remains additional places that critical data can live in the EHR that I suspect would not be picked up by an application such as this.

8. One most notable aspect of this research that was not highlighted is that all of the design work took place on/in a Cerner based environment. This is notable since most HCD and SMART-on-FHIR application development has been on EPIC implementations. This should be highlighted.

9. From my understanding, Indiana University Health is transitioning to EPIC in about 2 years. This should be mentioned and the authors should comment on whether it would affect RCT or implementation plans.

Reviewer #2: The review introduces a novel CDSS prototype based on SMART on FHIR that addresses the problem of scattered patient information across the patient record. However, there are a few areas which can be improved upon:

1. There is limited usability data and ideally validating it on the site’s EHR data would be the best approach but given that limitation, the claims should match accordingly. Currently, the claims are overreaching and not supported by the limited usability data.

2. Adding supplementary information so that the current logic can be rebuilt would also be great.

Reviewer #3: This manuscript describes the design and pilot evaluation of SMARTLiver, a SMART-on-FHIR–based clinical decision support system for cirrhosis management. The tool addresses a well-recognized problem fragmented EHR data and poor adherence to guidelines through human centered design and usability testing. The main value of the paper lies in showing that clinicians find data aggregation and workflow-sensitive CDS highly valuable, even without advanced AI. The study is timely, relevant, and aligns with the scope of PLOS Digital Health.

Comments

1. In the initial part of the Discussion section, the manuscript implies clinical readiness; this should be reframed to consistently present SMARTLiver as a prototype at the proof-of-concept stage, with clinical outcomes still to be demonstrated. Given the very small sample size (n=5), bold assertions about adoption or clinical impact should be avoided. Similarly, the statement “workflow integration trumps technical interoperability for clinical adoption” in the Discussion reads as a broad conclusion that cannot be supported from a pilot usability study

2. To set appropriate expectations from the outset, the title could also be adjusted to indicate that this is a pilot or early stage feasibility study.

**Do you want your identity to be public for this peer review?** For information about this choice, including consent withdrawal, please see our Privacy Policy

Reviewer #1: No

Reviewer #2: No

Reviewer #3: No

**Figure resubmission:**

**Reproducibility:** To enhance the reproducibility of your results, we recommend that authors of applicable studies deposit laboratory protocols in protocols.io, where a protocol can be assigned its own identifier (DOI) such that it can be cited independently in the future. Additionally, PLOS ONE offers an option to publish peer-reviewed clinical study protocols. Read more information on sharing protocols at https://plos.org/protocols?utm_medium=editorial-email&utm_source=authorletters&utm_campaign=protocols

---

## [Decision Letter · Decision Letter 1]

5 Jan 2026

A pilot feasibility study of human-centered design for cirrhosis care: Development and pilot testing of SMARTLiver prototype, a FHIR-based clinical decision support system for hepatology

PDIG-D-25-00499R1

Dear Dr. Purkayastha,

We are pleased to inform you that your manuscript 'A pilot feasibility study of human-centered design for cirrhosis care: Development and pilot testing of SMARTLiver prototype, a FHIR-based clinical decision support system for hepatology' has been provisionally accepted for publication in PLOS Digital Health.

Best regards,

Jasmit Shah, PhD

Guest Editor

PLOS Digital Health

**Additional Editor Comments (if provided):**

**Reviewer Comments (if any, and for reference):**

Reviewer's Responses to Questions

**Comments to the Author**

Reviewer #1: All comments have been addressed

Reviewer #2: All comments have been addressed

Reviewer #3: All comments have been addressed

publication criteria?

Reviewer #1: Yes

Reviewer #2: Yes

Reviewer #3: Yes

3. Has the statistical analysis been performed appropriately and rigorously?

Reviewer #1: Yes

Reviewer #2: Yes

Reviewer #3: Yes

4. Have the authors made all data underlying the findings in their manuscript fully available (please refer to the Data Availability Statement at the start of the manuscript PDF file)?

Reviewer #1: Yes

Reviewer #2: Yes

Reviewer #3: Yes

5. Is the manuscript presented in an intelligible fashion and written in standard English?

Reviewer #1: Yes

Reviewer #2: Yes

Reviewer #3: Yes

Reviewer #1: The authors have addressed all of my comments, I recommend acceptance.

Reviewer #2: The authors have substantially addressed my previous concerns. They have reframed the work as a pilot feasibility study of a prototype and the claims are now more appropriately aligned with the limited usability data and lack of live deployment. They have also added technical detail and referenced supplementary materials which should allow reconstruction of the core logic. Additionally, they can share the github link to the code repository to make it easier to reproduce.

Reviewer #3: Thank you for addressing my comments

**Do you want your identity to be public for this peer review?** For information about this choice, including consent withdrawal, please see our Privacy Policy

Reviewer #1: No

Reviewer #2: **Yes:** Shrey Lakhotia

Reviewer #3: No
